evolution, genetics, microbiology

kin recognition, adaptive heterogeneity, cooperation, *Saccharomyces cerevisiae*, adhesin, *FLO11*

**Author for correspondence:**
Helen A. Murphy
e-mail: hamurphy@wm.edu

†Present address: Department of Biology, University of Pennsylvania, Philadelphia, PA, USA.

# Variation at an adhesin locus suggests sociality in natural populations of the yeast *Saccharomyces cerevisiae*

Zachary J. Oppler†, Meadow E. Parrish and Helen A. Murphy

Department of Biology, William & Mary, PO Box 8795, Williamsburg, VA 23187-8795, USA

HAM, 0000-0002-4363-4543

Microbes engage in numerous social behaviours that are critical for survival and reproduction, and that require individuals to act as a collective. Various mechanisms ensure that collectives are composed of related, cooperating cells, thus allowing for the evolution and stability of these traits, and for selection to favour traits beneficial to the collective. Since microbes are difficult to observe directly, sociality in natural populations can instead be investigated using evolutionary genetic signatures, as social loci can be evolutionary hotspots. The budding yeast has been studied for over a century, yet little is known about its social behaviour in nature. Flo11 is a highly regulated cell adhesin required for most laboratory social phenotypes; studies suggest it may function in cell recognition and its heterogeneous expression may be adaptive for collectives such as biofilms. We investigated this locus and found positive selection in the areas implicated in cell–cell interaction, suggesting selection for kin discrimination. We also found balancing selection at an upstream activation site, suggesting selection on the level of variegated gene expression. Our results suggest this model yeast is surprisingly social in natural environments and is probably engaging in various forms of sociality. By using genomic data, this research provides a glimpse of otherwise unobservable interactions.

## 1. Introduction

Microbes are capable of functioning as collectives to engage in social behaviours ranging from swarming and foraging to producing fruiting bodies and highly differentiated biofilms [1]. These behaviours require individual cells to work together; due to the likely appearance of cheating phenotypes, relatedness among interacting individuals is usually high and the behaviours are favoured through kin selection [2]. Various mechanisms exist to ensure the maintenance of cooperation [3]. Some are passive, such as spatial structure with low or no dispersal, which generates patches of identical clones [4]. Others are more active, such as kin discrimination with a preference for close relatives [5] or discrimination among like kinds (i.e. greenbeard loci in which cooperators recognize each other regardless of overall relatedness [6]). Microbial collectives themselves may be targets of selection, producing bet-hedging traits such as population heterogeneity [7]. Selection at this higher level can ultimately lead to the evolution of multicellularity [8–10].

Whether for purging cheaters, controlling recognition or modulating the level of social expression, social genes can be hotspots for evolution in microbial populations [11–16]. Because natural conditions are often unknown and are difficult to recapitulate in the laboratory, using genomic signatures at social loci to infer which traits are under selection and what historic and current processes have shaped microbial populations can be a powerful approach to understanding sociality [15,16]. The research presented here focuses on the budding yeast, *Saccharomyces cerevisiae*, and a major social locus, *FLO11*.

Saccharomyces cerevisiae is a biomedical model organism that has been studied for over a century. The favoured laboratory strains are 'well behaved' and grow without adhering to surfaces or one another. Yet, as environmental isolates have been collected, it has become clear that social traits such as biofilm formation, flocculation, plastic adherence and invasive growth are highly variable and relatively common [17]. While certain social phenotypes can be induced in the laboratory [18], it remains unknown exactly how yeast sociality manifests in natural settings. Each of the laboratory social phenotypes requires proteins on the outer surface of the cell, known as adhesins or flocculins, for adherence [19]. The flocculin Flo1 has been shown to function as a green-beard during flocculation in liquid [20] by binding to mannan oligosaccharides in the cell wall of other cells, but not to other Flo1 proteins [21], and has not been implicated in other social phenotypes.

Another adhesin, Flo11, functions in cell–cell and cell–surface adhesion, and is required for all spatially structured social phenotypes [19,22]. Flo11p has been hypothesized to function in cell–cell recognition and has the general structure of other eukaryotic recognition proteins: membrane-associated with an extracellular domain in the immunoglobulin superfamily. The highly repetitive middle domain of Flo11p pushes the extracellular domain away from the cell [21], and length variation has been associated with various social phenotypes [23] and yeast chronological ageing [24].

These types of recognition proteins are found when discrimination among self/non-self is required. In microbes, they play roles such as slug formation in social amoeba [25], and in multicellular organisms they govern traits such as histocompatibility loci [26], dendrite avoidance [27] and tissue recognition in clonal invertebrates [28,29]. As with other discrimination proteins, Flo11p from one cell must interact with Flo11p from another, allowing for homotypic interactions. Recent work showing that two variants of the extracellular domain of Flo11p exhibit a preference for homophilic binding support the possibility of a self-recognition role for this protein [30].

FLO11 is regulated by one of the largest promoters in the yeast genome [31]. In this region, transcription factors from conserved signalling pathways converge [23,32], transcription of two long non-coding RNAs creates a toggle [33] and chromatin remodelling leads to epigenetic silencing [34,35]. This complex regulatory circuitry suggests that expression of FLO11, and therefore induction of sociality, is an important cellular decision. Indeed, regulated, facultative expression of cooperation can be a robust strategy in microbes [36]. FLO11 regulation creates expression heterogeneity in clonal cultures, which is likely to be adaptive, as variegated FLO11 expression has been shown to lead to increased biomass and space usage in nutrient-limiting environments [37].

Evidence is mounting for the importance of this social locus in S. cerevisiae, yet not much is known about its sequence variation. The highly repetitive nature of the gene has made it challenging to confidently ascribe variation with short-read sequencing data, the kind found in most population genomic studies. Intriguingly, FLO11 from two strains was found to differ by a 15-amino acid insert in the domain responsible for cell-to-cell adherence [30], suggesting the potential for abundant natural variation.

As FLO11 has been implicated in cell recognition and adaptive heterogeneity in the laboratory, we used population genetic variation to test whether there was a detectable signature of these phenomena in natural populations. We predicted that if Flo11p functions in self-recognition, there would be hyper-variable regions associated with the domains responsible for cell-to-cell interaction, as has been shown in other microbes [14]. And if the level of variegated expression plays a role in ecological competition, we hypothesized there would be significant regulatory variation.

Our analysis uncovered the predicted pattern: positive selection in the regions of the gene responsible for cell–cell interactions, and a signature of balancing selection in a region previously shown to contribute to epigenetic silencing. These results highlight the extent to which this important model organism is likely to be social in its natural environment, and further demonstrates the power of using evolutionary genetic signatures to infer microbial sociality.

## 2. Material and methods

### (a) Strain panel and phenotyping

A total of 78 strains representing a variety of niches and geographical locations from public and private collections [38–40] were used for analysis (electronic supplementary material, table S1). All phenotyping was done with homothallic diploid strains, as S. cerevisiae probably exists as a diploid in natural settings. Biofilms were inoculated with overnight YPD cultures (1% yeast extract, 2% peptone, 2% dextrose); four replicates were each scored by two researchers. Mats were inoculated with 2 µl on 0.3% agar low dextrose (0.1%) YPD (LD) 35 × 10 mm plates, sealed and incubated upright at 25°C for 10 days, then imaged and scored for complexity using a four-point scale (electronic supplementary material, figure S1) [17]. Complex colonies were inoculated using a 96-pin multi-blot replicator on OmniTrays containing 2% agar LD medium. Plates were sealed and incubated at 30°C for 6 days before imaging on an Epson Expression 11000 XL scanner and scored for complexity using a five-point scale (electronic supplementary material, figure S1) [41].

### (b) Sequencing and data processing

Genomic DNA was extracted and the FLO11 locus, which included approximately 3 kb upstream and approximately 1 kb downstream of the coding region, was amplified with iProof high-fidelity polymerase (BioRad). Cleaned amplicons were sent to the University of Georgia Genomics and Bioinformatics core for KAPA library preparation of each amplicon and paired-end 300 bp sequencing on an Illumina MiSeq platform; the amplicons were multiplexed with two whole yeast genome samples, which made up 20% of the reads. Data have been deposited in the Short Read Archive (https://www.ncbi.nlm.nih.gov/sra) under BioProject ID PRJNA556160.

Raw reads were processed to generate de novo assemblies using GENEIOUS 10.0.9R10. For each strain, the longest consensus contig was aligned to the S. cerevisiae reference genome to verify that it mapped to the FLO11 locus. Reads were mapped back to these contigs to resolve ambiguous sites. For most samples, the pipeline was unable to resolve the repetitive B domain. The contigs were trimmed and processed into four files: the upstream region, the downstream region, the A domain and the C domain. To resolve remaining ambiguous SNPS, reads were mapped to the aligned, trimmed consensus sequences with BWA [42] and SNPs were called with FREEBAYES [43]; high-frequency SNPs (greater than 0.8) were replaced in the consensus sequence. Mapped reads were also viewed in Integrative Genomics Viewer (IGV) to manually curate any residual ambiguous sites. Most strains were derived from single spores, and therefore were not expected to contain heterozygous sites.

The B domain was PCR amplified with Phusion high-fidelity polymerase; fragment analysis was conducted using an Agilent 2100 BioAnalyzer with an Agilent DNA 7500 Kit. Files with curated sequences and length variation are available from the Dryad Digital Repository: http://dx.doi.org/10.5061/dryad.0zpc866t5 [44].

### (c) Evolutionary analysis

The PAML program package [45] was used to discern patterns of selection. Using codeml, both fixed and random sites maximum-likelihood models of amino acid substitutions were used with the various phylogenetic trees and alignments; analyses were performed on both the complete gene and on segments defined by recombination points, as inferred by GARD [46]. Tajima's $D$ and $\pi$ were calculated using the PopGenome package in R [47]. All phylogenetic trees were generated using PhyML [48] in GENEIOUS, using the HKY85 substitution model and 1000 bootstraps, followed by generation of a consensus tree.

A literature search identified 42 other genes (electronic supplementary material, table S2) encoding cell-wall-associated or GPI-anchored proteins. Sequences were extracted from publicly available genomes for 54 of the 78 strains (electronic supplementary material, table S3), aligned using MAFFT [49], and the pairwise non-synonymous (dN) and synonymous rates (dS) were estimated using yn00 in PAML. The curated *FLO11* data for the same strains were analysed for comparison.

### (d) Strain generation for fluorescence assays

HMY58 (YJM1083), a homozygous diploid derived from a single spore of NRRL Y-10988, HMY394, a homozygous diploid derived from a single spore of YJM311, and HMY270 (YPS681) were transformed using a lithium acetate procedure with either an *mCherry-KanMX* or *GFP-KanMX* cassette targeted to the terminal region of *PGK1*. The original strains were also transformed with a *NatMX* or *GFP-KanMX* cassette targeted to the *FLO11* gene. Single spores were isolated from the transformed strains and mated to generate strains heterozygous for fluorescence and Δ*flo11*, as well as hybrids (electronic supplementary material, table S4). *FLO11* alleles were verified in all strains with Sanger sequencing. All primers are listed in electronic supplementary material, table S5.

### (e) Functional analysis of the *FLO11* alleles

Mats were generated as described and inoculated with mixed cultures containing either an equal volume of strains or a 1 : 10 ratio by volume. For all strains and combinations, three to five mats were generated per assay; the 1 : 1 assay was conducted twice and the 1 : 10 assay was conducted once.

### (f) prFLO11-GFP expression

Strains were grown to saturation for 24 h in YPD or LD supplemented with G418 in 96-well plates. Samples were imaged and the number of cells in light and GFP images was counted manually. For each strain and medium combination, 10–12 wells were analysed. The data were analysed using an ANOVA approach in JMP v. 11.2.0.

## 3. Results

In order to investigate the natural genetic variation of this cellular adhesin, we amplified, sequenced and generated de novo assemblies of the regulatory and coding regions of 78 environmental isolates that varied in their social phenotypes (electronic supplementary material, figure S1).

Flo11p has three domains [21]. The A domain is implicated in cell–cell binding and has an immunoglobulin-like core along with a fungal-specific region [50] (figure 1a,b). Next, the B domain is a highly glycosylated repetitive serine–threonine-rich middle. Recombination events lead to length variation, which can change the level of adherence [23]. Finally, the C domain contains a glycosylphosphatidylinositol (GPI)-anchor signal, which facilitates its linkage to the cell wall matrix [21].

In the majority of strains, the analysis resolved the unique sequence in the A and C domains, as well as the upstream and downstream regions, but not the repetitive B domain. The alleles from seven strains were compared to their published genomes [51], thus verifying that the observed variation was not a technical artefact. The length of the B domain was determined by fragment analysis (figure 1c). Our results showed lengths ranging from approximately 650 to 3100 bp and confirm the potential for varying levels of adherence segregating in nature [52,53]. Overall, sequence alignment uncovered a large amount of genetic variation at this locus (electronic supplementary material, figures S2 and S3).

### (a) Testing for patterns that support cell recognition
### (i) Variation in the coding region

The previously reported 15-amino-acid insert in the A domain occurred in approximately 30% of the strains, and contained variation within it. The distribution of synonymous and non-synonymous variation in the A and C domains of the protein (figure 1d) suggested that positive selection was restricted to the regions of the gene responsible for cell–cell interactions, as indicated by a dN/dS ratio greater than 1. To test whether selection in these regions was indeed different, the evolution of the gene was modelled using codeml in PAML [54] with the A and C domains combined (electronic supplementary material, table S6). Using fixed site models, which allow heterogeneous evolution in partitioned data, the more likely model was the one in which the regions associated with cell–cell interactions evolved separately with a dN/dS ratio of approximately 4.6, compared to 0.5 for the rest of the non-repetitive gene sequence ($\chi^2 = 47.51$, $p < 0.0001$). Next, using random sites models, which make no *a priori* assumptions about which amino acids are evolving under different types of selection, the more likely model contained a class of amino acids under positive selection ($\chi^2 = 221.95$, $p < 0.0001$). Bayesian analysis identified 16 significant codons as evolving under positive selection, 15 of which were in the A domain, with 7 in the regions associated with cell–cell interaction (electronic supplementary material, table S7). Since the significant results could be driven by the 15-amino-acid insert, the strains with and without the insert were analysed separately; the same models were still found to be significant (electronic supplementary material, tables S8 and S9).

The results were robust: analysis with the A domain alone, analysis with the A and C domains concatenated (described above) and analysis that included a part of the B domain that could be resolved and aligned, revealed the same models to be significant with the identity of the positively selected codons varying slightly. It is possible that some of the genetic variation was the result of recombination and not de novo mutations (as is assumed in the evolutionary models); therefore, the same three datasets were tested for a

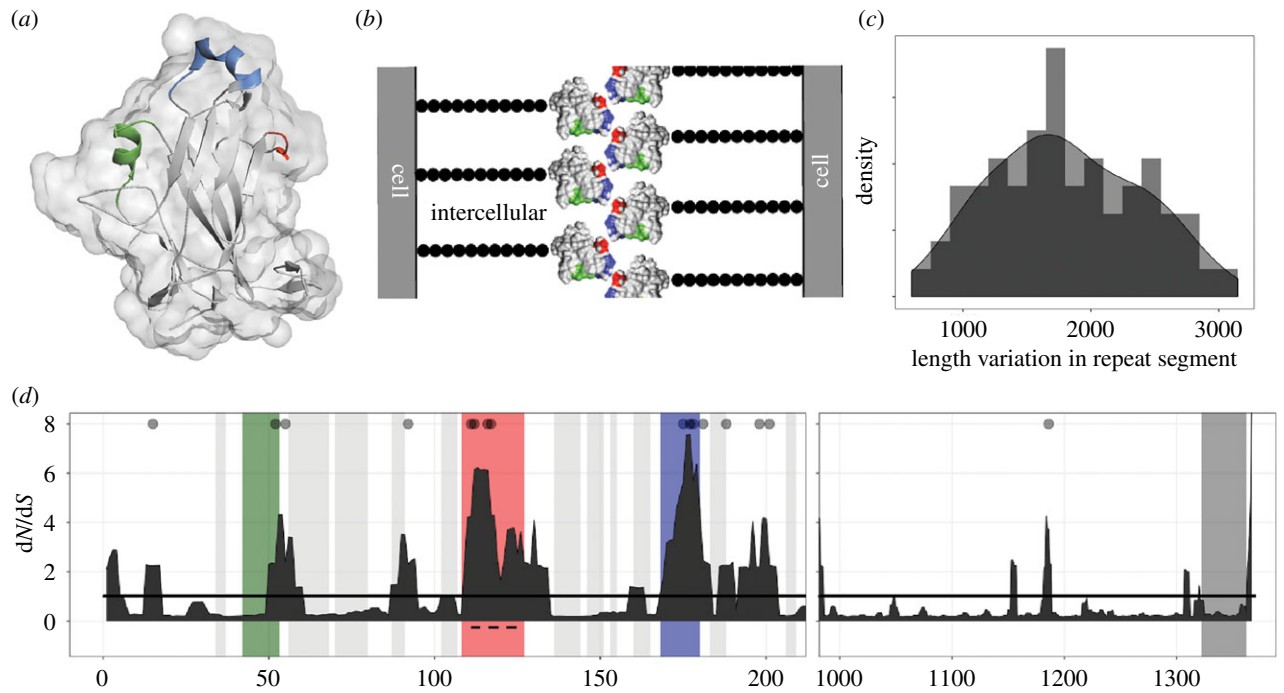

**Figure 1.** *FLO11* structure and coding genetic variation. (*a*) Structure of the A domain as reported in [50], generated in PyMol (Protein Data Bank: 4UYR). The green α-helix, blue $3_{10}$ α-helix, and the red $3_{10}$ turn comprise a fungal-specific region. (*b*) Schematic of a cell–cell interaction (not to scale): the A domain is hypothesized to interact through the blue and red highlighted features; black beads represent the B domain; the C domain is buried in the cell wall. (*c*) Frequency histogram of length variants of the B domain according to fragment analysis (*n* = 64 strains). (*d*) Plot of the d*N*/d*S* ratio of codons in the unique regions of the gene. *x*-axis: codon position in the A domain (left, *n* = 70 strains) and C domain (right, *n* = 76 strains); note the different scales. *y*-axis: five-codon sliding window of the posterior means of *ω* (d*N*/d*S*), as calculated in PAML using random sites model 2; grey dots indicate sites under significant positive selection ($p > 0.95$). The dashed line corresponds to the 15-amino-acid insert. (Online version in colour.)

signal of recombination using GARD. The analyses showed moderate support for a break point separating the two interacting regions in the A domain (although the precise location varied), as well as strong support for recombination within the repetitive B domain (electronic supplementary material, figure S4 and table S10). Separate analysis of the three recombination segments found the same models and codons to be significant as the analysis of the complete sequence (electronic supplementary material, tables S11 and S12).

Overall, we observed that the non-synonymous variation was clustered in the regions associated with cellular interactions and our analyses suggested diversifying selection in the precise regions required for cells to adhere to one another.

### (ii) Phylogenetic distribution of coding variation
A possible explanation for the observed pattern of diversity could be selection for different adhesion properties based on the ecological niche of the strain. However, when environmental origin and level of sociality were mapped onto the phylogenetic tree, there appeared to be no association (figure 2*a*). Furthermore, strains from the same geographical location and/or ecological niche exhibited genetic and phenotypic differences from one another. The panel included eight woodland isolates from three sites in Pennsylvania (designated with a YPS strain name), which probably inhabit the same niche; these strains exhibited variation in the portion of the gene responsible for cell–cell interaction. This observation supports the idea that there may be selection for recognition among interacting lineages. However, it should be noted that the ecological categories are broad and may not be able to capture informative fine-scale variation.

### (iii) Comparison to other genes
To determine whether the observed diversity at *FLO11* was unusually high, it was compared to the coding variation in other cell wall and GPI-anchored proteins (electronic supplementary material, figure S5). While *FLO11* had a higher overall d*N*/d*S* ratio, the individual d*N* and d*S* rates were comparable to the other proteins. This confirms that it is not the amount, but the location of the variation within the gene that is significant.

## (b) Testing for patterns that support adaptive expression
### (i) Variation in the regulatory region
Analysis of the regulatory sequences uncovered segregating variation (electronic supplementary material, figure S6) and positive values of Tajima's *D* at an upstream location (approx. −2800), which may indicate balancing selection [55] (figure 3*a*; electronic supplementary material, figure S7); in that area, three alleles were identified that contain eight linked SNPs. This particular region has been implicated in contributing to the expression 'toggle' at *FLO11* [33]. The histone deacetylase RpdL3 was shown to bind to this area, influencing both expression of ncRNAs and access of transcription machinery to *FLO11*.

### (ii) Phylogenetic distribution of regulatory variation
To test whether the variation was associated with niche and/or level of sociality, these traits were mapped onto the phylogenetic tree of the upstream region (figure 2*b*). The tree divided the strains into two main clades containing

Proc. R. Soc. B 286: 20191948

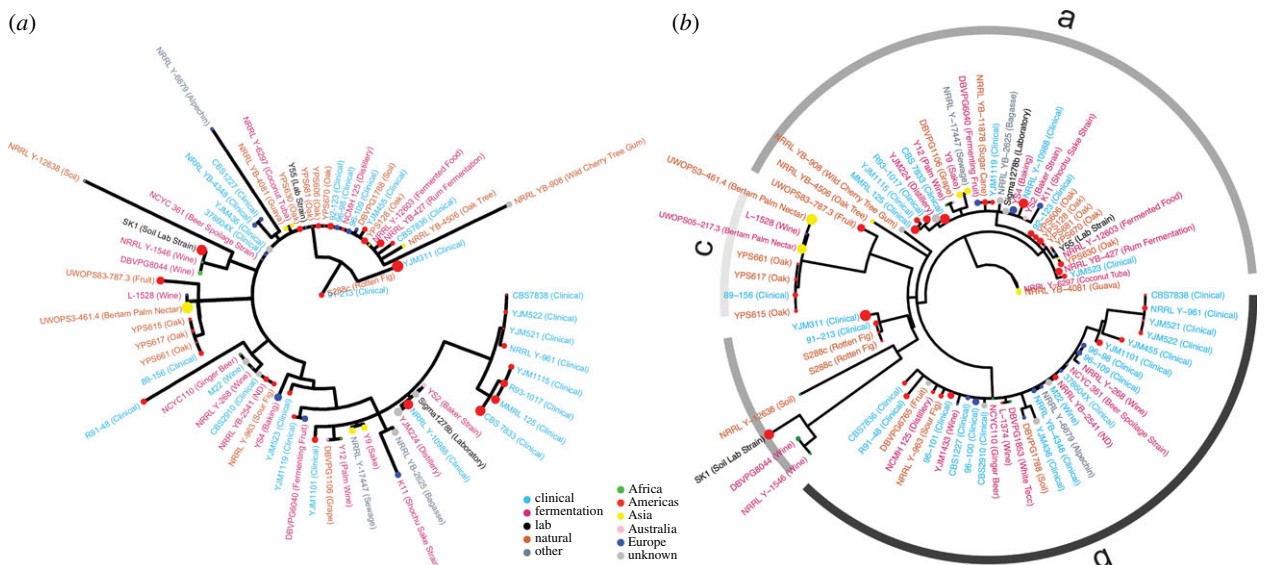

**Figure 2.** Phylogenetic trees of coding and regulatory regions show sociality is only associated with regulatory variation. Text colour indicates ecological niche, point colour indicates geographical origin and point size indicates the sum of the mat and biofilm-colony score. (*a*) Maximum-likelihood consensus tree of the alignment of the concatenated A and C domains. Black star indicates the 15-amino-acid insertion. (*b*) Maximum-likelihood consensus tree of the alignment of the upstream regulatory region; bars and letters indicate regulatory alleles at the site of balancing selection. (Online version in colour.)

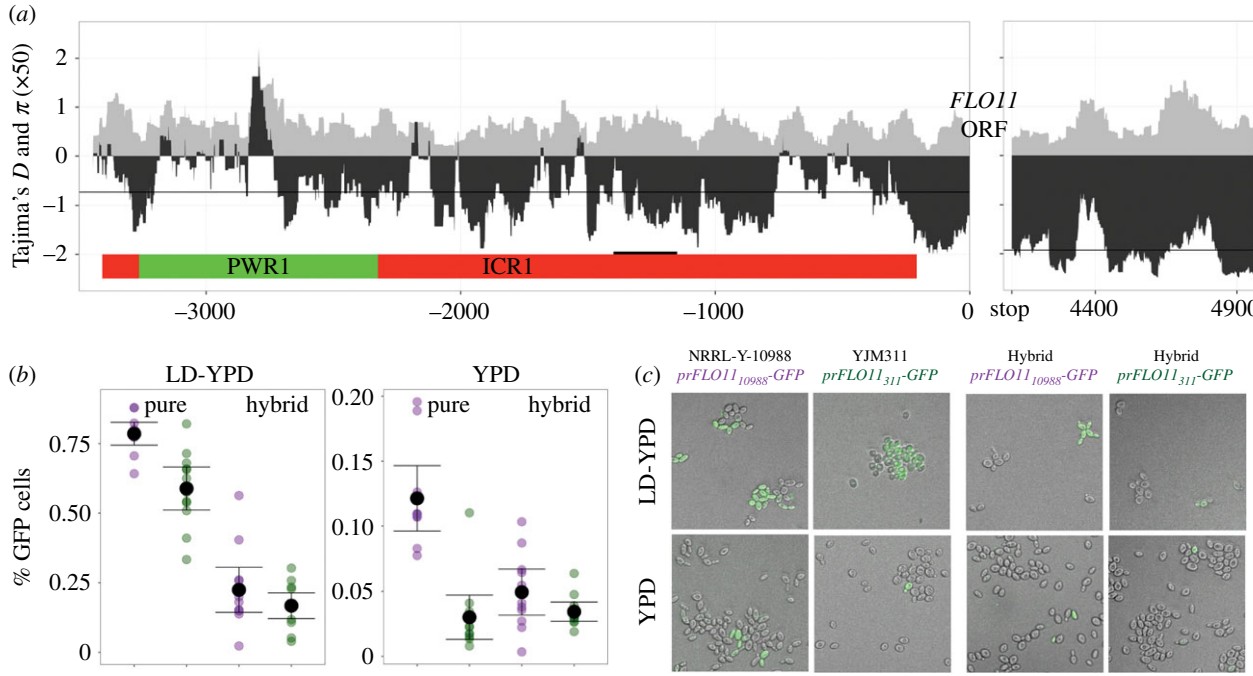

**Figure 3.** Location and effect of regulatory variation at the *FLO11* locus. (*a*) *x*-axis: nucleotide position of upstream and downstream regions of *FLO11*. Location of lncRNAs, *PWR1* (upregulates) and *ICR1* (downregulates) are depicted in coloured rectangles; black segment represents the binding site for major transcription factors Sfl1 and Flo8. *y*-axis: 100 bp sliding windows of nucleotide diversity ($\pi \times 50$, grey) and Tajima's *D* (black), for $n = 75$ and $n = 76$ strains; lines represent regional average of Tajima's *D*. (*b*) Purple and green points represent estimates of GFP expression from independent cultures of strains containing $prFLO11_{10988}$-*GFP* or $prFLO11_{311}$-*GFP*, respectively, each based on an image with an average of 110 and 445 cells in LD and YPD, respectively. Pure backgrounds are derived from original environmental isolates. Black points represent the strain average in a given medium ± 2 s.e.m. (*c*) Representative composite images of GFP expression. (Online version in colour.)

different alleles found in the region of balancing selection (with the exception of one laboratory strain, SK1). One clade contained the highly social strains and was overall, significantly more social ($p < 0.03$, *t*-test with unequal variances, electronic supplementary material, figure S8). Thus, it does appear that some regulatory variation is associated with the level of sociality. Given the range of phenotypes found within the more social clade, it is unlikely that the variation

at the site of balancing selection alone causes increased sociality; rather, there are probably modifiers selected within this allelic background.

While modifiers may exist anywhere within the genome, we tested whether there were any located within the regulatory region. In two social strains with the 'a' allele, but also separated by over 30 SNPs in the upstream region, one copy of *FLO11* was replaced with *GFP*. Hemizygous hybrid

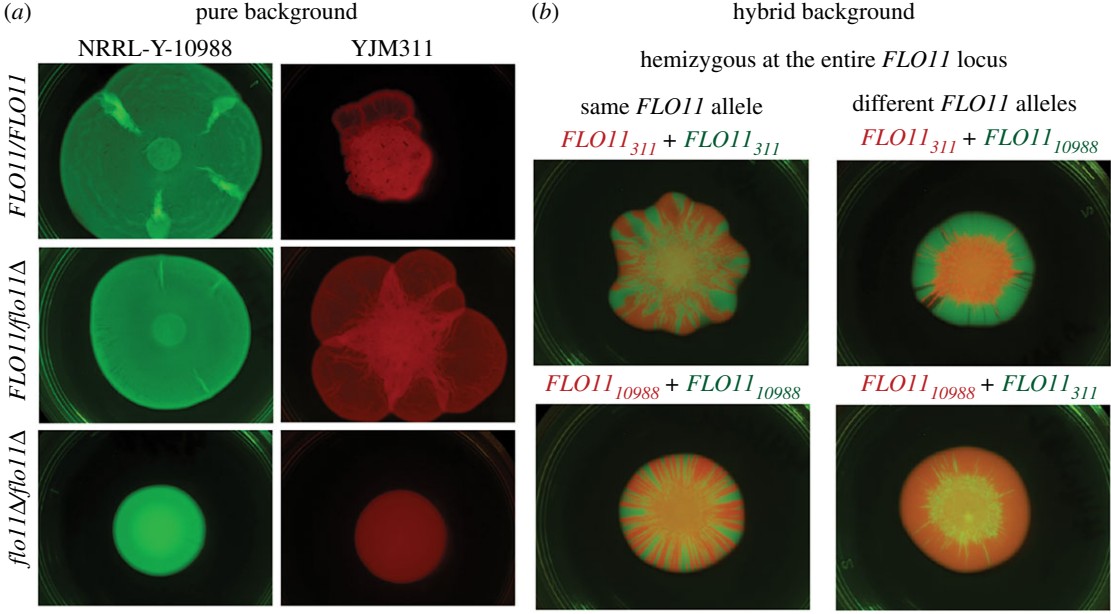

**Figure 4.** The effect of natural *FLO11* alleles. (*a*) Single-strain mats from environmental isolates with fluorescence markers. NRRL Y-10988 has the insert in the A-domain, while YJM311 does not; both strains have the A regulatory allele, but contain additional SNP regulatory variation. (*b*) Two-strain mats composed of hybrid strains with different fluorescent markers that were mixed 1 : 1 by volume before inoculation. Images for a third environmental background are in electronic supplementary material, figure S1. (Online version in colour.)

backgrounds were then generated with *GFP* under the control of one of the two different regulatory regions (figure 3*b*,*c*). The strains were grown to saturation and the percentage of GFP-expressing cells was estimated. GFP expression was different in the two original strains, with NRRL-Y-10988 showing greater expression than YJM311 ($p < 0.0001$; electronic supplementary material, table S13). In hybrid backgrounds, the trend of $prFLO11_{10988}$ having a higher expression level was maintained, although was not statistically significant ($p < 0.14$). This is consistent with modifiers of small, but detectable effect contributing to the level of sociality through regulatory variation. Low-dextrose conditions induced more GFP expression than rich medium ($p < 0.0001$). This result is not surprising, as glucose-limiting conditions have been shown to induce biofilm formation [56]; however, this observation supports the idea that the level of variegated expression may be selected under different environmental conditions.

## (c) Testing the effect of natural genetic variation on a laboratory social phenotype

To determine the effects of natural genetic variation at *FLO11*, the coding and regulatory variation should be isolated. However, due to the difficulty of genetically manipulating environmental strains, we were only able to analyse entire alleles (regulatory and coding together) in a limited number of backgrounds. Since social phenotypes have a complex genetic network underlying their expression [57], variants within these networks could influence the expression of *FLO11*. We therefore used hemizygous hybrids with high levels of genetic variation, and the effect of *FLO11* alleles was isolated in a background containing a large and constant set of possible trans-acting elements.

The effect of different *FLO11* alleles was assayed in mat biofilms, structured cooperative communities grown on viscous medium. In environmental isolates ranging in mat

expression, knocking out *FLO11* prevented biofilm formation (figure 4*a*; electronic supplementary material, figure S9), thus verifying previous reports of the role of this locus in social phenotypes [18].

To test whether naturally occurring alleles led to functional differences, hybrid strains with a single copy of *FLO11* were generated from three environmental isolates (figure 4*b*; electronic supplementary material, figure S9). There is clearly an effect of genetic background, as the hybrid biofilms are different than the original environmental backgrounds. However, there is also an effect of the *FLO11* allele: the same hybrid background generated different biofilm architectures depending on the identity of the *FLO11* allele. These mats were initiated with a 1 : 1 mix (by volume) of genetically identical strains differing only in fluorescence marker. The sectoring is indicative of genetic drift; the random placement of cells on the outer edge of the inoculum allows for the formation of sectors of descendants [58].

When mats were formed with two strains that *differed* at the *FLO11* locus, one allele appeared to outcompete the others. In microbial communities, monopolizing the edge of the community, which contains the available resources, allows genetic lineages to outcompete others [59]. In our assay, one of three alleles consistently monopolized the expanding front of the mat; these communities lack sectoring and instead have one dominant genotype. This result was robust to the identity of the fluorescence marker. We repeated the assay with the competitive allele at a numeric disadvantage in the inoculum (1 : 10 dilution by volume). The hybrid containing this allele was still able to reach and control all, or part, of the outer edge of the mat (electronic supplementary material, figure S9). Our results suggest that *FLO11* alleles may confer a competitive ability in particular ecological contexts. For example, the 'winning' allele in our assay could be favoured in naturally viscous environments, such as rotting fruits, but may not be favoured in other environments.

It is possible that the regulatory control of *FLO11*, the coding variation or a combination of both led to this 'winning' phenotype; however, the hemizygous assay was unable to distinguish among these possibilities. Importantly, while viscous laboratory medium lacks the intricacies of spatially structured micro-environments and may not recapitulate natural conditions, the results demonstrate that natural *FLO11* alleles can have a strong effect on a social phenotype.

## 4. Discussion

Microbes can be surprisingly social organisms, acting collectively to survive and reproduce in the face of innumerable biotic and abiotic hazards [1]. As such, social genes can be hotspots for evolution; they can be involved in a myriad of processes, including kin recognition, purging cheaters and modulating overall levels of sociality. Investigating the signature of these processes in the genome is a powerful approach to understanding microbial sociality. In the budding yeast, *FLO11* is one such locus, as variation in the regulatory and coding region has been implicated in important social and life-history traits. We used patterns of genetic variation at this locus to determine whether there was any evidence of sociality in natural populations, and to our knowledge, performed the first evolutionary analysis of the *FLO11*.

First, surprisingly, the results showed that the sites coding for the domains responsible for cell–cell interaction are under positive selection, thus implying the protein's role in self-discrimination. Since natural *S. cerevisiae* social expression is unknown, to test for recognition, the properties of the A domain should be isolated. Barua *et al.* [30] attached the adhesion domain of two alleles to magnetic beads and observed cell adherence. The results showed a preference for homophilic binding with heterotypic binding also possible. As the interaction between self-alleles was stronger than non-self, Flo11 appeared to have both greenbeard and self-recognition properties. In some microbial discrimination systems, rather than a binary system of cooperation or non-cooperation, the proteins responsible for cellular adhesion can exhibit a spectrum of cooperation (i.e. strength of cell-to-cell interaction) that is related to allelic variation, and is known as a 'polychromatic greenbeard'. The genetic results presented here support the existence of such a system in yeast, as the observed pattern of molecular variation is similar to that of Tgr [14], a polychromatic greenbeard in *Dictyostelium discoideum* [60].

Next, we amplified the repetitive B domain and found a large range of length variation, which indicates the possibility of varying levels of adherence and sociality in nature. Both length variation and the identity of the repeat units have been shown to alter the ability of yeast to engage in social behaviours with longer alleles tending towards stronger phenotypes [23]. In flor yeast, a longer B domain was shown to lead to a higher level of hydrophobicity, contributing to the ability to float [23], and high levels of length variation have been reported in flor-producing yeast [52,53]. Finally, in a recent mapping study, the length of this domain was implicated in chronological ageing, with the longer variant associated with a decrease in lifespan [24]. Intriguingly, coordinated programmed cell death is believed to be one of the steps necessary for the evolution of multicellularity [61].

Finally, analysis of the regulatory region uncovered balancing selection at a site involved in epigenetic regulation and variegated *FLO11* expression. Particular alleles were found associated with increased sociality, supporting the possibility of selection on heterogenous expression in clonal lineages. In mat biofilms, variegated expression leads to a faster spread across viscous agar and an increase in biomass [37]. In a growing, spatially structured community, individuals at the edge of the leading front have access to nutrients and space [58]; thus, production of substances that facilitate cell–cell adherence can be a competitive strategy [62,63]. Indeed, competitions between biofilm-forming and non-biofilm-forming yeast strains have shown a fitness benefit to biofilm production, with the biofilm-forming strains dominating the outer edge of the community [64]. We tested the effect of natural *FLO11* alleles and found that certain alleles were competitively dominant in mats, suggesting that the natural variation we observed can have a profound effect on social phenotypes.

The growing collection of environmental *S. cerevisiae* isolates [65] has shown that this yeast can be found in many different ecological niches—from deciduous woodlands to vineyards, dairy and fruit fermentations, and clinical settings—that numerous isolates have mixed genetic backgrounds, and dispersal occurs via insects [66], all of which suggest that yeast of different backgrounds encounter each other in the environment. This scenario allows for the possibility of both competition among lineages and selection for growth in different ecological niches. We hypothesize that inter-clonal competition drives the development of a recognition system, while abiotic factors select regulatory variants, or variants in the genetic network underlying expression level of social traits.

While *S. cerevisiae* has been associated with humans for thousands of years and has been studied extensively for over a century, little is known about its behaviour in the natural environment [67]. Our research used patterns of genetic variation at a social locus to provide a glimpse of otherwise unobservable interactions; our data support the hypothesis that this yeast is likely engaging in various forms of sociality. Future research in this system should systematically investigate the independent functional effects of the observed natural coding and regulatory variants.

Data accessibility. The phenotyping datasets and fasta files supporting this article have been uploaded at the Dryad Digital Repository: http://doi.org/10.5061/dryad.0zpc866t5 [44]. All raw sequences have been uploaded to SRA (https://www.ncbi.nlm.nih.gov/sra) under BioProject ID PRJNA556160.

Authors' contributions. Z.J.O. conducted the sequence and phylogenetic analyses, phenotyped strains, participated in the design of the study and co-wrote the manuscript. M.E.P. engineered and phenotyped strains. H.A.M. conceived of and designed the study, generated the sequence data, phenotyped strains and cowrote the manuscript. All authors gave final approval for publication and agree to be held accountable for the work performed therein.

Competing interests. We declare we have no competing interests.

Funding. The research was funded by a Jeffress Trust for Interdisciplinary Research Award and NIH grant no. R15-GM122032 to H.A.M., a William & Mary Charles Center Honors Fellowship to Z.J.O. and an HHMI Undergraduate Science Education Grant to William & Mary.

Acknowledgements. We thank Paul Magwene and Ed Louis for strains, Doug Young for help with the protein structure and Danting Jiang for computational assistance.

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
