## [Reviewer comments · Proceedings of the Royal Society B: Biological Sciences]

Review History

RSPB-2019-0736.R0 (Original submission)

Review form: Reviewer 1

Recommendation

Major revision is needed (please make suggestions in comments)

Scientific importance: Is the manuscript an original and important contribution to its field?

Good

General interest: Is the paper of sufficient general interest?

Good

Quality of the paper: Is the overall quality of the paper suitable?

Acceptable

Is the length of the paper justified?

Yes

Should the paper be seen by a specialist statistical reviewer?

No

Do you have any concerns about statistical analyses in this paper? If so, please specify them explicitly in your report.

Yes

It is a condition of publication that authors make their supporting data, code and materials available - either as supplementary material or hosted in an external repository. Please rate, if applicable, the supporting data on the following criteria.

Is it accessible?

Yes

Is it clear?

Yes

Is it adequate?

Yes

Do you have any ethical concerns with this paper?

No

Comments to the Author

Here I review the manuscript titled "Variation at an adhesin locus supports self-discrimination in the yeast *Saccharomyces cerevisiae*" by Zachary Oppler, Meadow Parrish and Helen Murphy which has been submitted to "Proceedings of the Royal Society B".

The authors present an interesting study that was motivated in the larger framework of kin and kind selection theory in microbes, for which they investigate the potential role of the adhesin FLO11 in cell-cell-adhesion interactions among many different strains of the budding yeast, *Saccharomyces cerevisiae*.

First, they describe the principle structure of the FLO11-peptide, which is implicated in biofilm formation in response to nutrient limitations. The authors draw analogies to the greenbeard locus FLO1, which is involved in flocculation and allows carriers of the same allele to form genetically heterogeneous cell-flocks. They present evidence from the literature for a potentially similar social function of Flo11, and set out to test if FLO11 ultimately shows signatures of selection and preferential binding which would allow to classify it as another possible green beard gene, too.

The authors present their results by introducing us to FLO11 which is partitioned into its three domains: A, B and C. The authors were able to sequence and assemble the coding sequences (cds) of the latter and the former for a set of ca. 80 natural budding yeast strains. The B-domain was much more elusive due to its highly repetitive nature, and thus they presented length-polymorphism data instead. They then went on to assess the cds for signals of selection using PAML with appropriate and careful parameter settings, and report strong signatures of positive selection in or near the relevant interaction domains, which convincingly argues for a role of this adhesin in cell-cell-interactions. Moreover, they demonstrate that balancing selection is shaping genetic diversity in the upstream PWR1 regulatory region of the FLO11-cds based on Tajima's D values. Finally, they ultimately test sets of divergent alleles in experimental biofilm formation assays using pairs of differently fluorescing hemizygous hybrids. They find evidence for preferential self-self-binding between FLO11-variants, which argues for a role in kin/kind

preference. There is also an apparent fitness difference between the tested alleles such that one allele (FLO11 (311) is “winning” in 1:1-mixes with FLO11 (10988)).

Zachary Oppler and colleagues present a strong case for evolution in an important biofilm-formation locus in the budding yeast based on a large sampling size, which set comprises ecologically highly diverse yeast isolates. This set allows the authors to comprehensively test the case for selective forces shaping the locus. While their results have potentially far-reaching implications, the study is somewhat hampered by the lack of strong experimental evidence, and a over-simplifying introduction to the field of kind/kin selection.

Major comments:

a) Assessing the role of recombination on signatures of selection in the coding regions

- The authors could not rule out that recombination could have shaped the strong patterns of selection they found. Since such a test exists and can be easily performed, I strongly recommend to apply this. The DataMonkey-Server (<https://www.datamonkey.org>) offers such a possibility: namely using the GARD method to detect recombination.
- Figure 1D: I don't understand the logic of the dashed line in the red region: it is extremely long, and seems to span the entire red part. If it indeed is the case that the 15AA-insertion spans most of that region, you need to reconsider your interpretation of positive selection based on a $dN/dS > 1$ (see comments on recombination above), what about running your analyses with and without the 15AA-residue insert?

b) Assessing the role of recombination on phylogenetic placement

The phylogenetic tree for FLO11 sequences is presented in Fig. 2. It appears that recombination could have strongly shaped this tree, as there is no clear pattern of groupings based on geography or ecology, plus there appears to be a 15-residue insertion that delineates 30% of the taxa into a single group. I wonder how the phylogeny would look based on conserved genes that would be representative of a “species tree”. Based on this, evidence for recombination could be assessed by comparing both phylogenies. I don't suggest the authors do need to generate new sequencing data, but suggest that if such data would exist on GenBank for at least some of the taxa, it would be worth comparing the FLO11-phylogenies from this subset of taxa with a tree derived from a set of conserved markers.

c) Assessing significance for non-neutral signatures.

I wonder if the authors could statistically rule out neutral evolution, ie, if they would be able to present a significance value for their Tajima's D evaluation presented in Figure 3 A. There is a way they could test this, namely by subdividing their data into the various up- and downstream regions presented in Fig. 3A and then perform the significance tests separately using Guillaume Achaz's `neutralitytest.c` as implemented here:

<http://www.abi.snv.jussieu.fr/achaz/neutralitytest.html>

d) Interpretation of the experimental biofilm assays.

While I find the author's argument convincing that FLO11 should be a factor mediating social interactions, I am not so convinced it is necessarily a greenbeard gene. It would be potentially more convincing if the authors could demonstrate the major importance that *flo11* plays in the observed social phenotypes and rule out that factors other than the FLO11-alleles determine the outcome of the biofilm assays. For example, the authors should integrate the *flo11*-null-mutant control data in the main results part, and stress how important FLO11 is for the biofilm formation. Moreover, FLO1 appears to be an important greenbeard gene in the budding yeast. Couldn't Flo1 could drive this observed pattern if it were in strong linkage disequilibrium with Flo11 for example?

e) Summary of the field.

I was not very pleased with how the authors introduce how microbes utilize “kin” and “kind” recognition for cooperative behaviours, and how evolution of these two phenomena seems to be solely driven by motility, which ultimately sets up their entire interest in performing their study. Non-motile cells also require discrimination systems, since even if non-motile, cells can be well-mixed in many environments. What about the concepts of “coming together” versus “staying together” type of evolution.

The authors state in lines 49 – 50: “In contrast, to motile microbes, many unicellular species live in spatially structured communities and grow clonally”. This reads as if motile organisms are not clonally propagating. Motile bacteria such as the myxobacteria also stay together in clonal patches while moving as a group, however, they are extremely kin discriminant. Or, take for example, the experimental evolution of snowflake yeasts (Ratcliff et al. 2011 PNAS), these yeasts were non-motile, but evolved a staying together phenotype that propagates together as a single unit. What about multicellular organisms consisting of myriads of nonmotile cells, they have strong self/non-self-recognition systems? I am also not convinced that we can, as of now, simply state that kind recognition is the most common phenomenon in microbes and hence that evolution in greenbeard genes is the norm. For example, the *traA*-gene in *M. xanthus* has not been demonstrated to be a greenbeard gene, as its role in social interactions in myxobacteria is extremely controversial (see Wielgoss et al. 2018 Mol Ecol). In a way, I have the feeling that the introduction is simplifying the field in a way that doesn’t do our current state of knowledge justice. I think the authors should revise both the abstract and introduction in light of these issues, and clarify what is generally accepted in the field.

Minor comments:

ll. 6-7: “[...] kin- and self-recognition have been reported; variable membrane-associated proteins confer discrimination.” The latter half-sentence appears to be a fragment, please revise.

l. 12: “Surprisingly,...” Don’t think this is appropriate wording here given that you suggest that you wanted to test for evidence supporting this hypothesis in the first place? Reconsider.

l. 90: “Variegated”: Please explain this term.

ll. 132-143 *Method: assembly.* Please explain, how your pipeline worked exactly: it is not entirely clear how you were able to get several kb of sequence from 300bp reads from single amplicons for each sequence in your set: eg, were amplicons sheared prior to loading, because if not, then you wouldn’t have got sequence variation? Moreover, you most likely needed a lot of spiked in phage DNA for your Illumina runs, in order to avoid signal overflow from uniform reads.

ll. 241 – 251 □ Please add a brief version of your interpretation of your phylogeny to the Figure 2’s caption, otherwise the reader looks at Fig. 2 and wonders what to look for (ie, tell the reader he needs to expect no specific pattern at all).

l. 256: “the individual rates”: Please clarify better, since you mean the different rates among FLO11 domains?

l. 624 [Fig 1 D]. For clarity: please provide a short summary of Fid. 1D before detailing the two axes.

Review form: Reviewer 2

Recommendation

Reject – article is scientifically unsound

Scientific importance: Is the manuscript an original and important contribution to its field?

Acceptable

General interest: Is the paper of sufficient general interest?

Good

Quality of the paper: Is the overall quality of the paper suitable?

Acceptable

Is the length of the paper justified?

Yes

Should the paper be seen by a specialist statistical reviewer?

No

Do you have any concerns about statistical analyses in this paper? If so, please specify them explicitly in your report.

No

It is a condition of publication that authors make their supporting data, code and materials available - either as supplementary material or hosted in an external repository. Please rate, if applicable, the supporting data on the following criteria.

Is it accessible?

Yes

Is it clear?

Yes

Is it adequate?

Yes

Do you have any ethical concerns with this paper?

No

Comments to the Author

The manuscript of Oppler et al examines the role of FLO11 in self recognition in yeast using two approaches: by analysis of the genetic variation in Flo11 and by creating strains carrying FLO11 alleles from environmental isolates (as hemizygous hybrid strains). They clearly demonstrate that certain domains of FLO11 that are involved in interactions are under direct selection for cell-cell recognition, while promoter regions (and possibly gene regulations) are influenced by environmental selection. In addition, the authors demonstrate that FLO11 alleles in hemizygous hybrid strains determine competition properties in spatially structured environment of colonies with assortment. They suggest that self-recognition plays an important role in non-motile microbes in addition to spatial segregation. The manuscript is clearly written and it presents an important context. However, their analysis is not connected to actual demonstration of self-recognition. The sequence analysis is not connected to experimental results, e.g. are there different groups of recognition? Do the strain that recognize themselves have more conserved or

similar domains? In the second part, introduction of new FLO11 alleles are proposed to determine self-recognition mediated segregation and fitness benefit, but is this due to specific regions in the FLO11 gene, or due to introduction of FLO11 alleles with certain specific properties? It was shown by Fidalgo et al (doi: 10.1073/pnas.0601713103) that rearrangement in the central tandem repeat domain alters evolutionary adaptation of *Saccharomyces* (in that case hydrophobicity). To be able to clearly state that the above predicted domains are responsible for the fitness benefit in these colonies, domain switch should be tested in the reintroduced alleles.

Decision letter (RSPB-2019-0736.R0)

30-Apr-2019

Dear Dr Murphy:

I am writing to inform you that your manuscript RSPB-2019-0736 entitled "Variation at an adhesin locus supports self-discrimination in the yeast *Saccharomyces cerevisiae*" has, in its current form, been rejected for publication in Proceedings B.

This action has been taken on the advice of referees, who have recommended that substantial revisions are necessary. With this in mind we would be willing to consider a resubmission, provided the comments of the referees are fully addressed. The conclusions from the Associate Editor and referees' reports are that the manuscript is too preliminary in its current state, but might possibly be publishable with more analysis and additional experiments. I should therefore emphasise that this is not a provisional acceptance, and if you feel you cannot address the requirements of the referees, I would advise submitting elsewhere.

Sincerely,
Loeske Kruuk
Editor
Proceedings B
mailto: proceedingsb@royalsociety.org

Associate Editor
Board Member: 1
Comments to Author:
Dear Dr Murphy

Your ms has been assessed by two expert reviewers. Both felt that the questions addressed in the ms are interesting and that you present some intriguing data. Several shortcomings were identified, however, that relate to: your treatment of the current literature for kin recognition in microbes; whether your data are sufficient to support your main conclusions; the sequence analysis; the lack of clear link between the sequence data and the experimental sections. Some additional experiments to test your key hypotheses are also suggested. I hope that these detailed reviewer comments are useful in revising your ms.

Yours sincerely
Mike

Reviewer(s)' Comments to Author:

Referee: 1

Comments to the Author(s)

Here I review the manuscript titled "Variation at an adhesin locus supports self-discrimination in the yeast *Saccharomyces cerevisiae*" by Zachary Oppler, Meadow Parrish and Helen Murphy which has been submitted to "Proceedings of the Royal Society B".

The authors present an interesting study that was motivated in the larger framework of kin and kind selection theory in microbes, for which they investigate the potential role of the adhesin FLO11 in cell-cell-adhesion interactions among many different strains of the budding yeast, *Saccharomyces cerevisiae*.

First, they describe the principle structure of the FLO11-peptide, which is implicated in biofilm formation in response to nutrient limitations. The authors draw analogies to the greenbeard locus FLO1, which is involved in flocculation and allows carriers of the same allele to form genetically heterogeneous cell-flocks. They present evidence from the literature for a potentially similar social function of Flo11, and set out to test if FLO11 ultimately shows signatures of selection and preferential binding which would allow to classify it as another possible green beard gene, too.

The authors present their results by introducing us to FLO11 which is partitioned into its three domains: A, B and C. The authors were able to sequence and assemble the coding sequences (cds) of the latter and the former for a set of ca. 80 natural budding yeast strains. The B-domain was much more elusive due to its highly repetitive nature, and thus they presented length-polymorphism data instead. They then went on to assess the cds for signals of selection using PAML with appropriate and careful parameter settings, and report strong signatures of positive selection in or near the relevant interaction domains, which convincingly argues for a role of this adhesin in cell-cell-interactions. Moreover, they demonstrate that balancing selection is shaping genetic diversity in the upstream PWR1 regulatory region of the FLO11-cds based on Tajima's D values. Finally, they ultimately test sets of divergent alleles in experimental biofilm formation assays using pairs of differently fluorescing hemizygous hybrids. They find evidence for preferential self-self-binding between FLO11-variants, which argues for a role in kin/kind

preference. There is also an apparent fitness difference between the tested alleles such that one allele (FLO11 (311) is “winning” in 1:1-mixes with FLO11 (10988)).

Zachary Oppler and colleagues present a strong case for evolution in an important biofilm-formation locus in the budding yeast based on a large sampling size, which set comprises ecologically highly diverse yeast isolates. This set allows the authors to comprehensively test the case for selective forces shaping the locus. While their results have potentially far-reaching implications, the study is somewhat hampered by the lack of strong experimental evidence, and an over-simplifying introduction to the field of kind/kin selection.

Major comments:

a) Assessing the role of recombination on signatures of selection in the coding regions

- The authors could not rule out that recombination could have shaped the strong patterns of selection they found. Since such a test exists and can be easily performed, I strongly recommend to apply this. The DataMonkey-Server (<https://www.datamonkey.org>) offers such a possibility: namely using the GARD method to detect recombination.
- Figure 1D: I don't understand the logic of the dashed line in the red region: it is extremely long, and seems to span the entire red part. If it indeed is the case that the 15AA-insertion spans most of that region, you need to reconsider your interpretation of positive selection based on a $dN/dS > 1$ (see comments on recombination above), what about running your analyses with and without the 15AA-residue insert?

b) Assessing the role of recombination on phylogenetic placement

The phylogenetic tree for FLO11 sequences is presented in Fig. 2. It appears that recombination could have strongly shaped this tree, as there is no clear pattern of groupings based on geography or ecology, plus there appears to be a 15-residue insertion that delineates 30% of the taxa into a single group. I wonder how the phylogeny would look based on conserved genes that would be representative of a “species tree”. Based on this, evidence for recombination could be assessed by comparing both phylogenies. I don't suggest the authors do need to generate new sequencing data, but suggest that if such data would exist on GenBank for at least some of the taxa, it would be worth comparing the FLO11-phylogenies from this subset of taxa with a tree derived from a set of conserved markers.

c) Assessing significance for non-neutral signatures.

I wonder if the authors could statistically rule out neutral evolution, ie, if they would be able to present a significance value for their Tajima's D evaluation presented in Figure 3 A. There is a way they could test this, namely by subdividing their data into the various up- and downstream regions presented in Fig. 3A and then perform the significance tests separately using Guillaume Achaz's `neutralitytest.c` as implemented here:

<http://www.abi.snv.jussieu.fr/achaz/neutralitytest.html>

d) Interpretation of the experimental biofilm assays.

While I find the author's argument convincing that FLO11 should be a factor mediating social interactions, I am not so convinced it is necessarily a greenbeard gene. It would be potentially more convincing if the authors could demonstrate the major importance that *flo11* plays in the observed social phenotypes and rule out that factors other than the FLO11-alleles determine the outcome of the biofilm assays. For example, the authors should integrate the *flo11*-null-mutant control data in the main results part, and stress how important FLO11 is for the biofilm formation. Moreover, FLO1 appears to be an important greenbeard gene in the budding yeast. Couldn't Flo1 could drive this observed pattern if it were in strong linkage disequilibrium with Flo11 for example?

e) Summary of the field.

I was not very pleased with how the authors introduce how microbes utilize “kin” and “kind” recognition for cooperative behaviours, and how evolution of these two phenomena seems to be solely driven by motility, which ultimately sets up their entire interest in performing their study. Non-motile cells also require discrimination systems, since even if non-motile, cells can be well-mixed in many environments. What about the concepts of “coming together” versus “staying together” type of evolution.

The authors state in lines 49 – 50: “In contrast, to motile microbes, many unicellular species live in spatially structured communities and grow clonally”. This reads as if motile organisms are not clonally propagating. Motile bacteria such as the myxobacteria also stay together in clonal patches while moving as a group, however, they are extremely kin discriminant. Or, take for example, the experimental evolution of snowflake yeasts (Ratcliff et al. 2011 PNAS), these yeasts were non-motile, but evolved a staying together phenotype that propagates together as a single unit. What about multicellular organisms consisting of myriads of nonmotile cells, they have strong self/non-self-recognition systems? I am also not convinced that we can, as of now, simply state that kind recognition is the most common phenomenon in microbes and hence that evolution in greenbeard genes is the norm. For example, the *traA*-gene in *M. xanthus* has not been demonstrated to be a greenbeard gene, as its role in social interactions in myxobacteria is extremely controversial (see Wielgoss et al. 2018 Mol Ecol). In a way, I have the feeling that the introduction is simplifying the field in a way that doesn’t do our current state of knowledge justice. I think the authors should revise both the abstract and introduction in light of these issues, and clarify what is generally accepted in the field.

Minor comments:

ll. 6-7: “[...] kin- and self-recognition have been reported; variable membrane-associated proteins confer discrimination.” The latter half-sentence appears to be a fragment, please revise.

l. 12: “Surprisingly,…” Don’t think this is appropriate wording here given that you suggest that you wanted to test for evidence supporting this hypothesis in the first place? Reconsider.

l. 90: “Variegated”: Please explain this term.

ll. 132-143 *Method: assembly.* Please explain, how your pipeline worked exactly: it is not entirely clear how you were able to get several kb of sequence from 300bp reads from single amplicons for each sequence in your set: eg, were amplicons sheared prior to loading, because if not, then you wouldn’t have got sequence variation? Moreover, you most likely needed a lot of spiked in phage DNA for your Illumina runs, in order to avoid signal overflow from uniform reads.

ll. 241 – 251 □ Please add a brief version of your interpretation of your phylogeny to the Figure 2’s caption, otherwise the reader looks at Fig. 2 and wonders what to look for (ie, tell the reader he needs to expect no specific pattern at all).

l. 256: “the individual rates”: Please clarify better, since you mean the different rates among FLO11 domains?

l. 624 [Fig 1 D]. For clarity: please provide a short summary of Fid. 1D before detailing the two axes.

Referee: 2

Comments to the Author(s)

The manuscript of Oppler et al examines the role of FLO11 in self recognition in yeast using two approaches: by analysis of the genetic variation in Flo11 and by creating strains carrying FLO11 alleles from environmental isolates (as hemizygous hybrid strains). They clearly demonstrate that certain domains of FLO11 that are involved in interactions are under direct selection for cell-cell recognition, while promoter regions (and possibly gene regulations) are influenced by environmental selection. In addition, the authors demonstrate that FLO11 alleles in hemizygous hybrid strains determine competition properties in spatially structured environment of colonies with assortment. They suggest that self-recognition plays an important role in non-motile microbes in addition to spatial segregation. The manuscript is clearly written and it presents an important context. However, their analysis is not connected to actual demonstration of self-recognition. The sequence analysis is not connected to experimental results, e.g. are there different groups of recognition? Do the strain that recognize themselves have more conserved or similar domains? In the second part, introduction of new FLO11 alleles are proposed to determine self-recognition mediated segregation and fitness benefit, but is this due to specific regions in the FLO11 gene, or due to introduction of FLO11 alleles with certain specific properties? It was shown by Fidalgo et al (doi: 10.1073/pnas.0601713103) that rearrangement in the central tandem repeat domain alters evolutionary adaptation of *Saccharomyces* (in that case hydrophobicity). To be able to clearly state that the above predicted domains are responsible for the fitness benefit in these colonies, domain switch should be tested in the reintroduced alleles.

Author's Response to Decision Letter for (RSPB-2019-0736.R0)

See Appendix A.

RSPB-2019-1948.R0

Review form: Reviewer 1 (Sebastien Wielgoss)

Recommendation

Accept as is

Scientific importance: Is the manuscript an original and important contribution to its field?

Good

General interest: Is the paper of sufficient general interest?

Good

Quality of the paper: Is the overall quality of the paper suitable?

Good

Is the length of the paper justified?

Yes

Should the paper be seen by a specialist statistical reviewer?

No

Do you have any concerns about statistical analyses in this paper? If so, please specify them explicitly in your report.

No

It is a condition of publication that authors make their supporting data, code and materials available - either as supplementary material or hosted in an external repository. Please rate, if applicable, the supporting data on the following criteria.

Is it accessible?

Yes

Is it clear?

Yes

Is it adequate?

Yes

Do you have any ethical concerns with this paper?

No

Comments to the Author

I was invited to comment on the revised manuscript by Oppler, Parrish and Murphy titled "Variation at an adhesin locus suggests sociality in natural populations of the yeast *Saccharomyces cerevisiae*".

I am quite pleased by the author's efforts to improve the manuscript in light of both reviewer's earlier comments on the first draft.

I would like to give a brief summary and then comment on the reviewer's response to my comments below.

The authors have focused on the population genetics assessment of, *flo11*, a gene encoding a potential social phenotype, mat formation. They analyzed the separate A,B, and C domains in great detail and using state-of-the-art methods. They find strong signatures of positive selection in domains that have been previously shown to affect cell-to-cell-interactions.

Finally, they also demonstrate experimentally that *flo11* is likely an important mediator of of sociality among natural yeast isolates. This is because functional *Flo11* is facilitating multicellular mat formation, but if the same gene is inactivated it removes this ability. When mixing differently labeled hybrid strains with variable genetic backgrounds, but identical *flo11*-alleles, random genetic drift in the microcolony leads to sector formation. This demonstrates, that the *flo11*-allele has equal strength in both (highly variable) genetic backgrounds. However, if hybrid backgrounds have each a different allele of *flo11*, one allele leads to a competitive advantage over the other, irrespective of genetic background. This demonstrates that conflicts arise when *flo11* is different, and this is settled by competition.

All in all, I am happy with the presented results and that the authors took their time to consider and often carry out approaches that can rule out artifactual signals of selection, such as recombination within the *flo11* gene.

I also want to point out, that when I meant testing the impact of “recombination” it was in each case “recombination within single genes” used to build the phylogeny. This is because, one of the most important aspects during phylogenetic inference is to take care of recombination within genes, as this leads to mixing different evolutionary histories and thereby compromising inferences made.

This is also, what I meant when suggesting that a comparison between “species tree” and “flo11-trees” (there are two in Fig.2). Such a comparison could help to identify any deviations from the genomic history of yeast strains the authors used for their study and compare this to the flo11 focal domains to visualize the potential impact of within-gene recombination. I think the authors have already shown the impact of such recombination in a different way, and that’s enough.

I am now convinced that the publication is good to go.

Review form: Reviewer 2

Recommendation

Accept as is

Scientific importance: Is the manuscript an original and important contribution to its field?

Good

General interest: Is the paper of sufficient general interest?

Excellent

Quality of the paper: Is the overall quality of the paper suitable?

Excellent

Is the length of the paper justified?

Yes

Should the paper be seen by a specialist statistical reviewer?

Yes

Do you have any concerns about statistical analyses in this paper? If so, please specify them explicitly in your report.

No

It is a condition of publication that authors make their supporting data, code and materials available - either as supplementary material or hosted in an external repository. Please rate, if applicable, the supporting data on the following criteria.

Is it accessible?

Yes

Is it clear?

Yes

Is it adequate?

Yes

Do you have any ethical concerns with this paper?

No

Comments to the Author

No further comments. Required changes were applied, although no experiments were added.

Decision letter (RSPB-2019-1948.R0)

17-Sep-2019

Dear Dr Murphy

I am pleased to inform you that your Review manuscript RSPB-2019-1948 entitled "Variation at an adhesin locus suggests sociality in natural populations of the yeast *Saccharomyces cerevisiae*" has been accepted for publication in Proceedings B.

The referees have not recommended any further changes. However referee 1 has made a couple of comments with regard to your response to reviewers, and I leave it up to you as to whether you wish to make any changes to your manuscript in relation to clarifying these issues. Therefore, please proof-read your manuscript carefully and upload your final files for publication. Because the schedule for publication is very tight, it is a condition of publication that you submit the revised version of your manuscript within 7 days. If you do not think you will be able to meet this date please let me know immediately.

To upload your manuscript, log into <http://mc.manuscriptcentral.com/prsb> and enter your Author Centre, where you will find your manuscript title listed under "Manuscripts with Decisions." Under "Actions," click on "Create a Revision." Your manuscript number has been appended to denote a revision.

You will be unable to make your revisions on the originally submitted version of the manuscript. Instead, upload a new version through your Author Centre.

- 1) A text file of the manuscript (doc, txt, rtf or tex), including the references, tables (including captions) and figure captions. Please remove any tracked changes from the text before submission. PDF files are not an accepted format for the "Main Document".
- 2) A separate electronic file of each figure (tiff, EPS or print-quality PDF preferred). The format should be produced directly from original creation package, or original software format. Please note that PowerPoint files are not accepted.
- 3) Electronic supplementary material: this should be contained in a separate file from the main text and the file name should contain the author's name and journal name, e.g `authorname_procb_ESM_figures.pdf`

All supplementary materials accompanying an accepted article will be treated as in their final form. They will be published alongside the paper on the journal website and posted on the online figshare repository. Files on figshare will be made available approximately one week before the

accompanying article so that the supplementary material can be attributed a unique DOI. Please see: <https://royalsociety.org/journals/authors/author-guidelines/>

4) Data-Sharing and data citation

It is a condition of publication that data supporting your paper are made available. Data should be made available either in the electronic supplementary material or through an appropriate repository. Details of how to access data should be included in your paper. Please see <https://royalsociety.org/journals/ethics-policies/data-sharing-mining/> for more details.

<http://datadryad.org/submit?journalID=RSPB&manu=RSPB-2019-1948> which will take you to your unique entry in the Dryad repository.

Once again, thank you for submitting your manuscript to Proceedings B and I look forward to receiving your final version. If you have any questions at all, please do not hesitate to get in touch.

Yours sincerely,

Professor Loeske Kruuk
<mailto:proceedingsb@royalsociety.org>

Reviewer(s)' Comments to Author:

Referee: 1

Comments to the Author(s).

I was invited to comment on the revised manuscript by Oppler, Parrish and Murphy titled "Variation at an adhesin locus suggests sociality in natural populations of the yeast *Saccharomyces cerevisiae*".

I am quite pleased by the author's efforts to improve the manuscript in light of both reviewer's earlier comments on the first draft.

I would like to give a brief summary and then comment on the reviewer's response to my comments below.

The authors have focused on the population genetics assessment of, *flo11*, a gene encoding a potential social phenotype, mat formation. They analyzed the separate A,B, and C domains in great detail and using state-of-the-art methods. They find strong signatures of positive selection in domains that have been previously shown to affect cell-to-cell-interactions.

Finally, they also demonstrate experimentally that flo11 is likely an important mediator of sociality among natural yeast isolates. This is because functional Flo11 is facilitating multicellular mat formation, but if the same gene is inactivated it removes this ability. When mixing differently labeled hybrid strains with variable genetic backgrounds, but identical flo11-alleles, random genetic drift in the microcolony leads to sector formation. This demonstrates, that the flo11-allele has equal strength in both (highly variable) genetic backgrounds. However, if hybrid backgrounds have each a different allele of flo11, one allele leads to a competitive advantage over the other, irrespective of genetic background. This demonstrates that conflicts arise when flo11 is different, and this is settled by competition.

All in all, I am happy with the presented results and that the authors took their time to consider and often carry out approaches that can rule out artifactual signals of selection, such as recombination within the flo11 gene.

I also want to point out, that when I meant testing the impact of "recombination" it was in each case "recombination within single genes" used to build the phylogeny. This is because, one of the most important aspects during phylogenetic inference is to take care of recombination within genes, as this leads to mixing different evolutionary histories and thereby compromising inferences made.

This is also, what I meant when suggesting that a comparison between "species tree" and "flo11-trees" (there are two in Fig.2). Such a comparison could help to identify any deviations from the genomic history of yeast strains the authors used for their study and compare this to the flo11 focal domains to visualize the potential impact of within-gene recombination. I think the authors have already shown the impact of such recombination in a different way, and that's enough.

I am now convinced that the publication is good to go.

Referee: 2

Comments to the Author(s).

No further comments. Required changes were applied, although no experiments were added.

Decision letter (RSPB-2019-1948.R1)

23-Sep-2019

Dear Dr Murphy

I am pleased to inform you that your manuscript entitled "Variation at an adhesin locus suggests sociality in natural populations of the yeast *Saccharomyces cerevisiae*" has been accepted for publication in Proceedings B.

Open Access

Paper charges

Sincerely,

Proceedings B

Appendix A

The College Of
WILLIAM & MARY

Helen Murphy, Assistant Professor
Department of Biology
P.O. Box 8795 / Williamsburg, VA 23187-8795
(757)-221-2216 / hamurphy@wm.edu

August 20, 2019

Dear Drs. Brockhurst and Kruuk,

We thank the Editorial team for sending out our manuscript, "Variation at an adhesin locus supports self-discrimination in the yeast *Saccharomyces cerevisiae*" for review by experts in the field. We appreciate the thorough and considered comments/suggestions from the reviewers. We found their comments to be constructive and feel that our manuscript has been significantly clarified and improved by making the changes they suggested.

We have copied the editor's and reviewers' comments below in italics and included our response to each. We look forward to hearing your response, and the response of the reviewers to our changes. Thank you again for your time and consideration.

Sincerely,

Helen Murphy

Summary of the Major Points and Associated Revisions

From the Associate Editor:

Both felt that the questions addressed in the ms are interesting and that you present some intriguing data. Several shortcomings were identified, however, that relate to: your treatment of the current literature for kin recognition in microbes; whether your data are sufficient to support your main conclusions; the sequence analysis; the lack of clear link between the sequence data and the experimental sections. Some additional experiments to test your key hypotheses are also suggested.

We have updated the overall inspiration and framing of the paper in accordance with the comments of both reviewers, and made substantial changes to the Abstract, Introduction, and Discussion. Our manuscript reports population genetic results for a major social locus in the model organism *S. cerevisiae*. While this organism is a biomedical model, has been studied for over a century, and is in all likelihood social, very little is known about its behavior in nature. Our results suggest that numerous types of sociality may be occurring, including recognition and adaptive heterogeneity, both of which have been demonstrated in the lab by other groups. While we include phenotyping results to demonstrate that the variation we observe can have profound functional consequences, we feel that those results are secondary to the population genetic ones. Since sociality in nature is a black box for this organism, our phenotyping results must be taken with a grain of salt. Rather, letting the population genetic signatures inform us is a powerful way to gain insight into the natural ecology of this model organism. Our initial submission focused on the population genetic results suggesting kin

recognition because that was the most exciting to us; however, we realize now that framing the manuscript in that way detracts from the larger message of the observed overall variability at this locus and what it implies. We have changed the title to reflect this new framing of the paper: Variation at an adhesin locus suggests sociality in natural populations of the yeast *Saccharomyces cerevisiae*

Aside from changing the framework for the manuscript, we have conducted the sequence analyses recommended by Reviewer 1. We have also conducted further phylogenetic analysis of the regulatory region. Finally, we have attempted the experiment proposed by Reviewer 2 innumerable times to no avail. We recognize that its absence is a shortcoming of one of the conclusions of our paper; however, we believe that the current framing of the manuscript demonstrates the importance of the findings at this social locus (hypervariable regions in the interacting domains, length variation in the gene, and balancing selection of 3 alleles in the regulatory region). Given the extraordinary implications of this variability and the significance of this model organism, we still believe that our results merit publication. We hope you agree.

Referee: 1

... Zacchary Oppler and colleagues present a strong case for evolution in an important biofilm-formation locus in the budding yeast based on a large sampling size, which set comprises ecologically highly diverse yeast isolates. This set allows the authors to comprehensively test the case for selective forces shaping the locus. While their results have potentially far-reaching implications, the study is somewhat hampered by the lack of strong experimental evidence, and an over-simplifying introduction to the field of kind/kin selection.

Major comments:

- a) *Assessing the role of recombination on signatures of selection in the coding regions*
- *The authors could not rule out that recombination could have shaped the strong patterns of selection they found. Since such a test exists and can be easily performed, I strongly recommend to apply this. The DataMonkey-Server (<https://www.datamonkey.org>) offers such a possibility: namely using the GARD method to detect recombination.*

We found this suggestion very helpful and included the analysis in the manuscript (lines 232-240, Tables S6-8, Figure S4). To summarize, there was strong support for a recombination breakpoint in the B domain, which is not surprising, given the repetitive nature of that section of the gene. There was modest support for a breakpoint separating the two interacting regions in the A-domain (corresponding to the blue and red in all the figures); subsequent independent PAML analysis of the three recombinant segments confirmed the originally reported results (i.e., the cell-cell interacting regions are under positive selection with numerous codons appearing significant).

- *Figure 1D: I don't understand the logic of the dashed line in the red region: it is extremely long, and seems to span the entire red part. If it indeed is the case that the 15AA-insertion spans most of that region, you need to reconsider your interpretation of positive selection based on a $dN/dS > 1$ (see comments on recombination above), what about running your analyses with and without the 15AA-residue insert?*

The 15-AA insertion adds to a region that is otherwise 4-AA long. We followed the suggestion of the reviewer and divided the data set into strains with and without the insert. Subsequent analysis of each set of strains supported the original findings (lines 225-227, Tables S4-5), including that one of the amino acids in the 4-AA region was under positive selection in the strains without the insert.

b) *Assessing the role of recombination on phylogenetic placement*

The phylogenetic tree for FLO11 sequences is presented in Fig. 2. It appears that recombination could have strongly shaped this tree, as there is no clear pattern of groupings based on geography or ecology, plus there appears to be a 15-residue insertion that delineates 30% of the taxa into a single group. I wonder how the phylogeny would look based on conserved genes that would be representative of a "species tree". Based on this, evidence for recombination could be assessed by comparing both phylogenies. I don't suggest the authors do need to generate new sequencing data, but suggest that if such data would exist on GenBank for at least some of the taxa, it would be worth comparing the FLO11-phylogenies from this subset of taxa with a tree derived from a set of conserved markers.

This is a reasonable suggestion, but we are not sure that including a phylogenetic tree of the conserved genes would provide new information (we had already made a tree with ~40 GPI-anchored and cell wall proteins for one of the PAML analyses that we originally presented; see Table S2 and S3). Most of the strains we are using are from the 100 Genomes Collection and a phylogeny of all those strains is presented in the original paper (Strope, et al., 2015, *Genome Research*, 10.1101/gr.185538.114). More importantly, the global phylogeny of yeast strains is well documented (Liti, et al., 2009, *Nature*, 10.1038/nature07743; Peters, et al., 2019, *Nature*, 10.1038/s41586-018-0030-5) and the results are robust: there are clusters from particular geographic regions, as well as many strains that are intermediates due to outcrossing (known in the yeast field as "mosaics"). Thus, including such a tree for this subsample of the global collection seems superfluous to us. The reviewer suggests that it is a good way to test for recombination; however, in sexual organisms, recombination is assumed and a single gene tree usually does not reflect the "species" or "strain" trees. The interesting thing in our case is to ask whether selection is favoring certain alleles to be associated with certain niches, which would show up in a phylogeny of the gene matching the ecology, and which is what we present in the manuscript. (Of course, it turns out that it does not for the coding region, but for the regulatory variation, the tree is associated with level of sociality.)

c) *Assessing significance for non-neutral signatures.*

I wonder if the authors could statistically rule out neutral evolution, ie, if they would be able to present a significance value for their Tajima's D evaluation presented in Figure 3 A. There is a way they could test this, namely by subdividing their data into the various up- and downstream regions presented in Fig. 3A and then perform the significance tests separately using Guillaume Achaz's neutralitytest.c as implemented here: <http://www.wabi.snv.jussieu.fr/achaz/neutralitytest.html>

We appreciate this suggestion and investigated using the neutrality test cited by the reviewer. This test calculates a single T-D value for the entire region inputted into the program. We contemplated dividing the regulatory regions into sections that our sliding-window analysis highlighted as potentially interesting, but since there was no *a priori* reason to suspect the region associated with the binding site for RpdL3 would be under balancing selection, it seemed that such a neutrality test would be invalid. Rather, we followed the procedure that many genomic studies employ, namely, to compare values of T-D of small regions (sliding window) to that of the local region and look for outliers. These distributions are now presented in Figure S7. This approach simply highlights sites that may be of interest for further functional analysis, which is what we performed for Figure 3.

d) *Interpretation of the experimental biofilm assays.*

While I find the author's argument convincing that FLO11 should be a factor mediating social interactions, I am not so convinced it is necessarily a greenbeard gene. It would be potentially more convincing if the authors could demonstrate the major importance that flo11 plays in the observed social phenotypes and rule out that factors other than the FLO11-alleles determine the outcome of the biofilm assays. For example, the authors should integrate the flo11-null-mutant control data in the

main results part, and stress how important FLO11 is for the biofilm formation. Moreover, FLO1 appears to be an important greenbeard gene in the budding yeast. Couldn't Flo1 could drive this observed pattern if it were in strong linkage disequilibrium with Flo11 for example?

While we wholeheartedly agree that it is important to demonstrate that *FLO11* is required for biofilm formation, we are building on work that has previously shown this to be true (e.g., Reynolds and Fink, 2001, *Science*, 10.1126/science.291.5505.878) and don't want to suggest our work is the first to show it. Indeed, we point to at least one review that summarizes experimental data showing that *FLO11* is required for all spatially-structured social phenotypes (Verstrepen and Klis, 2006, *Mol. Microbiol.*, 10.1111/j.1365-2958.2006.05072.x). We therefore include the *flo11* null images to verify that it is also required in our strains specifically, but do not want to focus too much on this result, since it is already well established.

In terms of the question of whether allelic identity determined the outcome of the biofilm assays, we appreciate the reviewer questioning the result and realized we needed to be clearer in the manuscript; we have updated the language to reflect this confusion. To answer the question, the only difference between the hybrid strains is the identity of the *FLO11* locus and the fluorescence/antibiotic marker (hemizygous hybrid assays are common in yeast genetic research: Ehrenreich, et al., 2010, *Nature*, 10.1038/nature08923). Since we see the same results when the fluorescence/antibiotic marker is reversed, we can conclude that it is due to the *FLO11* locus. But as noted in the manuscript, we cannot determine if it is regulation or the identity of the protein leading to the difference in mat formation. We do point to a published study showing that *FLO11* expression level can have profound effects on mat biofilm formation (Regenberg, et al., 2016, *Proc B*, 10.1098/rspb.2016.1303), which may point to regulation being foremost in our observed "winner".

The reviewer is correct that Flo1 has been suggested to be a greenbeard in *S. cerevisiae*, specifically in flocculation in liquid (Smukalla, et al., 2008, *Cell*, 10.1016/j.cell.2008.09.037), but it has not been implicated in any other phenotypes. In fact, the expression of all the *FLO* adhesin genes was investigated in mat biofilm formation, and only *FLO11* was present (Reynolds, et al. 2008, *Eukaryot. Cell*, 10.1128/EC.00310-06). There is also unlikely to be any linkage, as they are on separate chromosomes.

e) *Summary of the field.*

I was not very pleased with how the authors introduce how microbes utilize "kin" and "kind" recognition for cooperative behaviours, and how evolution of these two phenomena seems to be solely driven by motility, which ultimately sets up their entire interest in performing their study. Non-motile cells also require discrimination systems, since even if non-motile, cells can be well-mixed in many environments. What about the concepts of "coming together" versus "staying together" type of evolution.

The authors state in lines 49 – 50: "In contrast, to motile microbes, many unicellular species live in spatially structured communities and grow clonally". This reads as if motile organisms are not clonally propagating. Motile bacteria such as the myxobacteria also stay together in clonal patches while moving as a group, however, they are extremely kin discriminant. Or, take for example, the experimental evolution of snowflake yeasts (Ratcliff et al. 2011 PNAS), these yeasts were non-motile, but evolved a staying together phenotype that propagates together as a single unit. What about multicellular organisms consisting of myriads of nonmotile cells, they have strong self/non-self-recognition systems? I am also not convinced that we can, as of now, simply state that kind recognition is the most common phenomenon in microbes and hence that evolution in greenbeard genes is the norm. For example, the traA-gene in M. xanthus has not been demonstrated to be a greenbeard gene, as its role in social interactions in myxobacteria is extremely controversial (see Wielgoss et al. 2018 Mol Ecol). In a way, I have the feeling that the introduction is simplifying the field

in a way that doesn't do our current state of knowledge justice. I think the authors should revise both the abstract and introduction in light of these issues, and clarify what is generally accepted in the field.

We appreciate the spirit of this comment, but rather than responding in detail, we simply note that we have changed the framework of the Abstract, Introduction, and Discussion substantially and hope that it is satisfactory. We believe that it better represents the field and the relevance of our approach.

Minor comments:

ll. 6-7: "[...] kin- and self-recognition have been reported; variable membrane-associated proteins confer discrimination." The latter half-sentence appears to be a fragment, please revise.

l. 12: "Surprisingly,..." Don't think this is appropriate wording here given that you suggest that you wanted to test for evidence supporting this hypothesis in the first place? Reconsider.

l. 90: "Variegated": Please explain this term.

l. 256: "the individual rates": Please clarify better, since you mean the different rates among FLO11 domains?

We have considered and changed some of the wording pointed out by the reviewer.

*ll. 132-143 *Method: assembly.* Please explain, how your pipeline worked exactly: it is not entirely clear how you were able to get several kb of sequence from 300bp reads from single amplicons for each sequence in your set: eg, were amplicons sheared prior to loading, because if not, then you wouldn't have got sequence variation? Moreover, you most likely needed a lot of spiked in phage DNA for your Illumina runs, in order to avoid signal overflow from uniform reads.*

We had a single library made for each PCR product; the library preparation included fragmentation. We multiplexed these libraries with 2 whole genome samples, rather than spiking with phage DNA. We now include this information in the Methods section (lines 124-127).

ll. 241 – 251 ∠ Please add a brief version of your interpretation of your phylogeny to the Figure 2's caption, otherwise the reader looks at Fig. 2 and wonders what to look for (ie, tell the reader he needs to expect no specific pattern at all).

We changed the title of the figure to summarize the result and added a second panel for the regulatory results.

l. 624 [Fig 1 D]. For clarity: please provide a short summary of Fid. 1D before detailing the two axes.

Done.

Referee: 2

The manuscript of Oppler et al examines the role of FLO11 in self recognition in yeast using two approaches: by analysis of the genetic variation in Flo11 and by creating strains carrying FLO11 alleles from environmental isolates (as hemizygous hybrid strains). They clearly demonstrate that certain domains of FLO11 that are involved in interactions are under direct selection for cell-cell recognition, while promoter regions (and possibly gene regulations) are influenced by environmental selection. In addition, the authors demonstrate that FLO11 alleles in hemizygous hybrid strains determine competition properties in spatially structured environment of colonies with assortment.

They suggest that self-recognition plays an important role in non-motile microbes in addition to spatial segregation. The manuscript is clearly written and it presents an important context. However, their analysis is not connected to actual demonstration of self-recognition. The sequence analysis is not connected to experimental results, e.g. are there different groups of recognition? Do the strain that recognize themselves have more conserved or similar domains?

In the second part, introduction of new FLO11 alleles are proposed to determine self-recognition mediated segregation and fitness benefit, but is this due to specific regions in the FLO11 gene, or due to introduction of FLO11 alleles with certain specific properties? It was shown by Fidalgo et al (doi: 10.1073/pnas.0601713103) that rearrangement in the central tandem repeat domain alters evolutionary adaptation of Saccharomyces (in that case hydrophobicity). To be able to clearly state that the above predicted domains are responsible for the fitness benefit in these colonies, domain switch should be tested in the reintroduced alleles.

We agree that the ideal test of the recognition hypothesis is switching the A domain among strains. We have spent the better part of a year attempting this experiment with a CRISPR system, but have run into many technical difficulties working with the environmental isolates. A bigger issue is that it is unclear what true recognition would look like in a biofilm mat (larger sectioning?). Further complicating this issue is that we do not know what social phenotypes most yeast are expressing in nature. We are currently pursuing other more appropriate approaches, including a form of the bead adherence assay performed by another group and referenced in the manuscript, but this has also proved challenging. At the current time, we do not have these experiments working, although we continue to pursue them.

We agree that in the second part of the manuscript, in which we present biofilm assays, the results could be due to the protein itself (recognition, hydrophobicity, etc.) or even the regulation of the gene; these possibilities are acknowledged in the manuscript. What we clearly show, however, is that the identity of the allele at the locus can have large effects on a lab social phenotype. We hope that by reframing our paper, the implications of the extraordinary amount of variation at this social locus, and the functional effect of different alleles, will convince the reviewer of the merit of our work.